# Physical Function Tests Are Potential Tools to Identify Low Physical Resilience in Women after Breast Cancer Treatment

Fernanda Maria Martins [1], Anselmo Alves de Oliveira [1], Gersiel Oliveira-Júnior [1], Marcelo A. S. Carneiro [1,2], Luís Ronan Marquez Ferreira de Souza [1,3], Vitor Carvalho Lara [3], Rosekeila Simões Nomelini [1,4], Cláudio Oliveira Assumpção [1,5], Markus Vinícius Campos Souza [1,5] and Fábio Lera Orsatti [1,5,*]

1    Applied Physiology, Nutrition and Exercise Research Group (PhyNEr), Exercise Biology Research Lab (BioEx), Federal University of Triangulo Mineiro, Uberaba 38025-180, Brazil

2    Metabolism, Nutrition and Exercise Laboratory, Physical Education and Sport Center, Londrina State University, Londrina 86057-970, Brazil

3    Department of Radiology and Diagnostic Imaging, Federal University of Triangulo Mineiro, Uberaba 38025-180, Brazil

4    Department of Gynecology and Obstetrics, Federal University of Triangulo Mineiro, Uberaba 38025-180, Brazil

5    Department of Sport Sciences, Health Science Institute, Federal University of Triangulo Mineiro, Uberaba 38025-180, Brazil

\*    Correspondence: fabio.orsatti@uftm.edu.br

**Abstract:** Background: This study sought to investigate whether different physical function tests (objective measures of physical performance) may identify a low physical resilience in breast cancer survivors (BCS). Methods: This analytical cross-sectional study evaluated 146 BCS and 69 age-matched women without breast cancer history. The different times after the end of treatment were used as criteria for group division. Participants were divided into four groups: control (CT: n = 69–women without breast cancer history); <1.0 years after the end of treatment (<1 YAT: n = 60); 1–3.9 years after the end of treatment (1–3.9 YAT: n = 45); and ≥4 years after the end of treatment (>4 YAT: n = 41). Physical function was evaluated by 4 m walk test (4-MWT), five-times-sit-to-stand test (FTSST), timed up and go test (TUG), and short physical performance battery (SPPB). Age, menopausal status, smoking, number of medications, level of physical activity, body mass index, and muscle strength were used as confounding variables in ANCOVA. Results: All groups that underwent cancer treatment (<1 YAT, 1–3.9 YAT and ≥4 YAT) had lower physical performance ($p < 0.001$) identified by 4 MWT, TUG, and FTSST when compared to the CT group. For the SPPB, the <1 YAT and ≥4 YAT groups had lower performance ($p = 0.005$) when compared to the CT. Conclusions: The different physical function tests can be used to identify a low physical resilience in BCS.

**Keywords:** resilience; cancer; physical performance; clinical environment; physical performance



## 1. Introduction

Breast cancer accounts for 29.5% of all new cancer cases in Brazil [1]. More women are surviving breast cancer as a result of medical advances and early detection [2,3]. These women, however, continue to experience adverse effects after being cured, negatively affecting their quality of life and survival time [4,5]. One of the most concerning findings is that cancer survivors have a lower physical function (or physical performance), which is linked to higher mortality rates [5].

Breast cancer treatment (i.e., chemotherapy, radiotherapy, and hormone therapy) has been associated with the impairment of several homeostatic systems, such as multihormonal dysregulation [6], an increased inflammatory profile [7], changes in the central and peripheral nervous systems [8], a decline in energy reserve capacity [9], and fatigue [9]. As a result of multiple physiological systems being compromised, patients are more likely

to exhibit impairments in muscle strength and physical function [10], increasing their mortality risk [5].

Physical resilience is defined as the ability of an individual to resist or recover from a decline in physical function following exposure to a health stressor. (e.g., breast cancer treatment) [11,12]. Physical resilience has been understood as a dynamic process of interaction between physiological systems in an attempt to re-establish the original balance, maintain the initial phenotypic identity of the system, and preserve the system's functionality after the action of a stressor [11–15]. Based on the proposal of a physical resilience model, physiological reserve and homeostatic systems are associated with the physical resilience capacity [11,14]. In this regard, low physical resilience is the result of wear (i.e., deregulation) and tear of the physiological reserve and homeostatic systems that are not able to restore the original balance and maintain the functionality of the organic system [15]. Consequently, low physical resilience is associated with comorbidities [13], lower quality of life [16], and higher risk of death when exposed to agent stressors [14]. Thus, consecutive evaluations of physical function following cancer treatment can provide health professionals with information regarding survivors' physical resilience, health, and prognosis.

Most studies have used self-reported measures of physical function, such as the Short-Form 36 or the European Organization for Research and Treatment of Cancer Quality of Life Questionnaire [17,18]. Self-reported measurement tools include benefits such as the ability to test a large number of people as well as being inexpensive and practical. However, because of the influence of interfering factors such as educational level, memory impairment, dishonest answers, omitted questions, and survey weariness, self-reported measurement tools may provide less reliable information about physical function [19–21]. In this regard, more appropriate measures to assess low physical resilience are required [11].

Objective measures of physical function (e.g., walking speed, timed up and go, and sit-to-stand tests) might be promising tools to identify cancer survivors with low physical resilience. Physical function tests can be used as indicators of changes in homeostatic systems and physiological reserve [10,22,23] and as a tool to distinguish functional age from chronological age [11,24]. Moreover, the physical function test has been shown to be a prognostic biomarker of mortality, low quality of life, and treatment-related complications among cancer survivors [5,25,26]. Furthermore, from a hierarchical standpoint, changes in muscle function (i.e., muscle strength) influence changes in physical performance [27]. Therefore, muscle strength tests may help determine the reason for changes in physical capacity following cancer treatment (i.e., a decline in muscle strength). Furthermore, physical function tests are quick, inexpensive, and easy-to-perform tools that do not require special equipment or training when compared to clinical tests [28,29]. In addition, they are applicable in the clinical environment [28] and have high reliability between evaluators and test/retest [30,31]. However, the effectiveness of different physical function tests to identify cancer survivors at risk of low physical resilience remains unclear. Therefore, this study aimed to investigate whether different physical function tests may identify a lower physical resilience in BCS.

## 2. Results

There were no statistical differences ($p > 0.05$) between four groups (CT, <1 YAT, 1–3.9 YAT, and ≥4 YAT) for general characteristics, marital status, education attainment, physical activity status and use of medicine. Moreover, there were no statistical differences ($p > 0.05$) between the BCS groups (<1 YAT, 1–3.9 YAT, and ≥4 YAT) for cancer treatment characteristics, except for end-of-treatment (years) (Table 1).

**Table 1.** Comparison of general characteristics, marital status, education attainment, physical activity status, cancer treatment and medical treatment between groups.

| | CT (n = 69) | <1 YAT (n = 60) | 1–3.9 YAT (n = 45) | ≥4 YAT (n = 41) | p |
|---|---|---|---|---|---|
| **General characteristics** | | | | | |
| Age (years) | 58.5 (56.6; 60.4) | 57.1 (54.9; 59.3) | 55.9 (53.2; 58.6) | 59.4 (56.9; 61.9) | 0.216 |
| BMI (kg/m$^2$) | 27.9 (26.6; 29.2) | 28.5 (27.2; 29.9) | 26.9 (24.9; 29.0) | 27.2 (26.1; 28.3) | 0.406 |
| Menopause (%) | 94.2 (n = 65) | 83.3 (n = 50) | 86.6 (n = 39) | 100.0 (n = 41) | 0.302 |
| Smoker (%) | 13.0 (n = 9) | 11.6 (n = 7) | 17.7 (n = 8) | 2.4 (n = 1) | 0.263 |
| **Marital status** | | | | | |
| Single (%) | 17.3 (n = 12) | 18.3 (n = 11) | 17.7 (n = 13) | 9.7 (n = 4) | 0.933 |
| Married (%) | 42.0 (n = 29) | 48.3 (n = 29) | 44.4 (n = 20) | 53.6 (n = 22) | 0.321 |
| Divorced (%) | 15.9 (n = 11) | 20.0 (n = 12) | 11.1 (n = 5) | 17.0 (n = 7) | 0.977 |
| Widow (%) | 24.6 (n = 17) | 13.3 (n = 8) | 15.5 (n = 7) | 19.5 (n = 8) | 0.373 |
| **Education attainment** | | | | | |
| Complete elementary school (%) | 53.6 (n = 37) | 51.6 (n = 31) | 53.3 (n = 24) | 60.9 (n = 25) | 0.751 |
| Complete high school (%) | 34.7 (n = 24) | 25.0 (n = 15) | 28.8 (n = 13) | 19.5 (n = 8) | 0.337 |
| College graduated (%) | 11.5 (n = 8) | 23.3 (n = 14) | 17.7 (n = 8) | 19.5 (n = 8) | 0.329 |
| **Physical activity status** | | | | | |
| Physical activity level (min/Week) | 808.8 (609.4; 1008.2) | 699.1 (504.6; 893.6) | 649.5 (487.8; 811.8) | 663.4 (467.6; 859.2) | 0.648 |
| Active (%) | 69.5 (n = 48) | 66.6 (n = 40) | 68.8 (n = 31) | 65.8 (n = 27) | 0.155 |
| Walking (min/Week) | 43.9 (17.4; 37.0) | 27.2 (17.4; 37.0) | 32.5 (19.8; 45.1) | 43.9 (29.2; 58.6) | 0.266 |
| Walking (%) | 75.3 (n = 52) | 65.0 (n = 39) | 71.1 (n = 32) | 73.1 (n = 30) | 0.862 |
| Moderate physical activity (min/Week) | 104.8 (77.4; 132.2) | 84.2 (60.9; 107.5) | 122.2 (74.5; 149.8) | 91.2 (64.5; 117.9) | 0.533 |
| Moderate physical activity (%) | 78.2 (n = 54) | 76.6 (n = 46) | 82.2 (n = 37) | 85.3 (n = 35) | 0.310 |
| Vigorous physical activity (%) | 17.3 (n = 12) | 15.0 (n = 9) | 13.3 (n = 6) | 17.0 (n = 7) | 0.843 |
| **Treatment of cancer ¥** | | | | | |
| Surgery (%) | ****** | 100.0 (n = 60) | 100.0 (n = 45) | 100.0 (n = 41) | 1.000 |
| Quadrantectomy surgical (%) | ****** | 50.0 (n = 30) | 62.2 (n = 28) | 48.7 (n = 20) | 0.440 |
| Mastectomy surgical (%) | ****** | 1.6 (n = 1) | 0.0 (n = 0) | 2.4 (n = 1) | 0.822 |
| Quadrantectomy and axillary dissection (%) | ****** | 8.3 (n = 5) | 8.8 (n = 4) | 7.3 (n = 3) | 0.914 |
| Mastectomy and axillary dissection (%) | ****** | 3.3 (n = 2) | 2.2 (n = 1) | 4.8 (n = 2) | 0.247 |
| Chemotherapy (session) | ****** | 31.4 (30.3; 32.6) | 31.8 (29.2; 34.4) | 29.6 (27.3; 32.0) | 0.434 |
| Radiotherapy (session) | ****** | 8.8 (7.5; 10.0) | 8.3 (7.2; 9.3) | 7.6 (6.3; 8.9) | 0.348 |
| Chemotherapy (n) | ****** | 10.0 (n = 6) | 4.4 (n = 2) | 19.5 (n = 8) | 0.193 |
| Radiotherapy (n) | ****** | 28.3 (n = 17) | 35.5 (n = 16) | 34.1 (n = 14) | 0.502 |
| Radiotherapy and Chemotherapy (n) | ****** | 61.6 (n = 37) | 60.0 (n = 27) | 46.3 (n = 19) | 0.145 |
| End-of-treatment (years) | ****** | 0.7 (0.6; 0.9) | 2.75 (2.5; 2.9) [#] | 7.02 (6.1; 7.8) [# †] | <0.001 |
| **Medical treatment** | | | | | |
| Depression drug (%) | 17.3 (n = 12) | 28.3 (n = 17) | 26.6 (n = 12) | 29.2 (n = 12) | 0.166 |
| Hypertension drug (%) | 33.3 (n = 23) | 35.0 (n = 21) | 33.3 (n = 15) | 39.0 (n = 16) | 0.626 |
| Diabetes drug (%) | 8.6 (n = 6) | 6.6 (n = 4) | 15.5 (n = 7) | 7.3 (n = 3) | 0.773 |

**Table 1.** *Cont.*

| | CT (n = 69) | <1 YAT (n = 60) | 1–3.9 YAT (n = 45) | ≥4 YAT (n = 41) | *p* |
|---|---|---|---|---|---|
| Thyroid drug (%) | 11.5 (n = 8) | 8.3 (n = 5) | 6.6 (n = 3) | 12.1 (n = 5) | 0.890 |
| Hypercholesterolemia drug (%) | 21.7 (n = 15) | 21.6(n = 13) | 15.5 (n = 7) | 26.8 (n = 11) | 0.643 |
| Tamoxifen drug ¥ (%) | ****** | 41.6 (n = 25) | 51.1 (n = 23) | 60.9 (n = 25) | 0.057 |
| Anastrazole drug ¥ (%) | ****** | 16.6 (n = 10) | 26.6 (n = 12) | 17.0 (n = 7) | 0.394 |
| Number of medications (n) | 1.8 (1.3; 2.2) | 1.8 (1.3; 2.3) | 1.9 (1.3; 2.4) | 2.0 (1.5; 2.6) | 0.845 |

CT = control group; <1 YAT = <1 year after the end of treatment group; 1–3.9 YAT = 1–3.9 years after the end of treatment group; ≥4 YAT = ≥4 years after the end of treatment group; BMI = body mass index; # = significantly different from <1 YAT; † = significantly different from 1–3.9 YAT; ¥ = the control group was excluded from the statistical analysis. Active (%) = the percentage of people who meet physical activity guideline/cut-off (at least 150 min of mild physical activity per week and/or at least 75 min of strenuous physical activity per week). Moderate physical activity (%) = the percentage of people who have engaged in moderate physical activity, regardless of the overall amount of physical activity they performed. Vigorous physical activity (%) = the percentage of people who have engaged in vigorous physical activity, regardless of the overall amount of physical activity they performed.

The parameters of physical function tests are shown in Table 2. After adjusting for confounding variables, YAT groups that underwent cancer treatment (<1 YAT, 1–3.9 YAT, and ≥4 YAT) showed lower physical performance in 4 MWT, TUG, and FTSST when compared to CT. The <1 YAT group and ≥4 YAT groups showed lower performance in SPPB when compared to CT. Only the 1–3.9 YAT group showed a lower MS value when compared to CT.

**Table 2.** Comparison of physical function between groups.

| | CT (n = 69) | <1 YAT (n = 60) | 1–3.9 YAT (n = 45) | ≥4 YAT (n = 41) | *p* | $\eta^2 p$ | OP |
|---|---|---|---|---|---|---|---|
| Unadjusted Values | | | | | | | |
| 4-MWT (m.s) | 1.2 (1.1; 1.3) | 1.1 (1.0; 1.1) * | 1.1 (1.0; 1.1) * | 1.0 (1.0; 1.1) * | 0.001 | 0.08 | 0.96 |
| FTSST (s) | 8.7 (7.9; 9.5) | 12.1 (11.4; 12.9) * | 11.4 (10.5; 12.4) * | 11.2 (10.3; 12.0) * | <0.001 | 0.17 | 1.00 |
| SPPB (score) | 11.4 (11.2; 11.7) | 10.9 (10.5; 11.2) * | 11.1 (10.8; 11.4) | 10.7 (10.2; 11.2) * | 0.023 | 0.04 | 0.75 |
| TUG (s) | 6.9 (6.5; 7.4) | 8.6 (8.2; 9.1) * | 8.4 (7.9; 9.0) * | 8.2 (7.6; 8.8) * | <0.001 | 0.13 | 0.99 |
| MS (kg) | 25.5 (24.1; 26.9) | 24.1 (22.6; 25.6) | 22.7 (21.1; 24.4) * | 25.8 (24.0; 27.5) | 0.035 | 0.03 | 0.67 |
| Adjusted Values | | | | | | | |
| 4-MWT (m.s) | 1.2 (1.2; 1.3) | 1.1 (1.0; 1.2) * | 1.1 (1.0; 1.2) * | 1.0 (0.9; 1.1) * | <0.001 | 0.13 | 0.99 |
| FTSST (s) | 8.6 (7.9; 9.3) | 11.9 (11.1; 12.7) * | 11.1 (10.2; 12.0) * | 11.8 (10.8; 12.7) * | <0.001 | 0.21 | 1.00 |
| SPPB (score) | 11.5 (11.2; 11.8) | 10.9 (10.6; 11.2) * | 11.2 (10.8; 11.5) | 10.8 (10.5; 11.2) * | 0.005 | 0.07 | 0.87 |
| TUG (s) | 6.9 (6.6; 7.3) | 8.4 (8.0; 8.9) * | 8.2 (7.7; 8.7) * | 8.3 (7.8; 8.8) * | <0.001 | 0.16 | 0.99 |
| MS (kg) | 25.8 (24. 4; 27.1) | 23.7 (22.2; 25.2) | 22.6 (20.9; 24.3) * | 26.7 (25.0; 28.5) † | 0.003 | 0.07 | 0.89 |

CT = control group; <1 YAT = <1 year after the end of treatment group; 1–3.9 YAT = 1–3.9 years after the end of treatment group; ≥4 YAT = ≥4 years after the end of treatment group; BMI = body mass index; 4-MWT = 4 m walk test; FTSST = five-times-sit-to-stand test; TUG = timed up and go test; SPPB = short physical performance battery; MS = muscle strength; * = significantly different from CT; † = significantly different from 1–3.9 YAT; ANCOVA adjusted for age, menopause, smoking, number of medications, BMI and level of physical activity, and MS (except for MS analysis); $\eta^2 p$ = partial eta-squared; OP = observed power.

## 3. Discussion

We found that physical function assessments are promising techniques for detecting low physical resilience in BSC. As longitudinal and long-term studies are difficult and expensive to conduct, this cross-sectional study adds to the field by offering new findings about prospective techniques for measuring physical resilience in BCS (i.e., oncology hospitals). Furthermore, these simple and low-cost techniques could be utilized by health professionals (e.g., doctors or nurses) to swiftly screen for low physical resilience in clinical settings. Early detection of inadequate physical resilience may lead to preventative

strategies aimed at individuals and populations. This is a significant discovery because inadequate physical resilience is associated with poor quality of life [16] and an increased risk of death [14] in cancer patients.

In the current study, physical function measured by 4 MWT, FTSST, TUG, and SPPB was lower in the BCS (i.e., the <1 YAT, 1–3.9 YAT, and ≥4 YAT groups) when compared to the age-matched women with no history of cancer (i.e., the CT group) (Table 2). These findings indicate that there is a reduction in physical function in the BCS and that this impairment does not recover even after a long period of time (4 years) after treatment ends. Physical function tests are widely used to assess physical health in different contexts [27,32]. For instance, physical function tests have been used to assess lower limb function [33], aerobic capacity [34], mobility risk [35], fall risk [36,37], disabilities [38], and balance dysfunction [39] in older adults, as well as being used as a prognostic biomarker of mortality and low quality of life among cancer survivors [5,25,26]. It has also been shown that physical function tests can be used as indicators of changes in physiological reserve and homeostatic systems in older adults [10,22,23]. In this regard, physiological reserve and homeostatic systems are related with a system's resilience capacity; that is, physical resilience is limited in part by the underlying physiological reserve in organ systems, according to the idea of a physical resilience model [10,11,14]. Thus, physical function tests may provide a measure of an individual's physiological reserve and, as a result, an indication of physical resilience in BSC in this construct.

We found that all physical performance tests were sensitive to identifying low physical resilience in BCS, regardless of the number of medications, muscle strength, menopause, smoking, BMI, and physical activity level (Table 2). Indeed, physical performance tests have been associated with a series of critical conditions and may reflect the decline in function of different physiological and homeostatic systems, such as body composition, homeostatic dysregulation, the energy system, and nervous system [10]. In terms of body composition, a decrease in muscle mass and, consequently, strength, as well as an increase in fat mass, result in a higher mechanical load that the patient must carry. As a result, skeletal muscles are always working at or near their maximum capacity (i.e., critical limit). When muscle strength decreases below the critical threshold, this is reflected as reduced physical capacity (i.e., reduced walking speed) [10]. Although we did not assess the change in body composition, BCS have higher body fat [40] and lower muscle strength [41] than age-matched women with no cancer treatment history. Another relevant point is that the impairment of various homeostatic systems, such as multihormonal dysregulation (i.e., levels of androgens, estrogens, estradiol, IGF-1, and insulin) and increased inflammatory profile related to cancer treatment [6,7], has been associated with the decline in mobility [10]. In addition, it has been shown that cancer treatment-associated changes in the nervous system [8], such as loss of myelinated and unmyelinated nerve fibers, demyelination partially compensated by remyelination, axonal atrophy, muscle fiber denervation, and altered autonomic responses, are associated with reduced performance in physical tests [10]. Additionally, several studies have shown that physical performance tests may mirror the energy reserve capacity [10,22,23,42,43]. In this regard, cancer treatment has been associated with changes in energy reserve capacity and fatigue [9]. Hence, changes in physical performance tests may be an indicator of a decline in the function of different physiological systems (e.g., energetic, hormonal, neural, and homeostatic) in BSC and, therefore, an expression of the capacity for physical resilience. Moreover, physical performance tests are an excellent predictor of declining physical function [44] and adverse health outcomes, such as cancer [5], cardiovascular disease [45], dementia [46], postoperative complications [47], and mortality in older adults and older BCS [48,49]. Therefore, our findings indicate that physical performance tests are potential tools capable of identifying low physical function and, consequently, physical resilience in BCS.

The physical performance tests are rarely incorporated as a standard assessment protocol in routine clinical practice. In that regard, the application of physical performance tests in the clinical setting can provide a rich resource for oncological medicine. In this

perspective, our study indicates that physical performance tests can provide a sensitive and objective measure of the capacity for physical resilience and, consequently, allow physicians to establish priorities in the care plan and avoid future complications to patients' health. Recently, it has been shown that identifying the dynamic process of physical resilience requires multiple repeated measurements of physical function (physical function recovery trajectories) after a stressor [50]. Thus, physical performance tests are quick application tools and result in different physical function responses [28]. It can be suggested that physical performance tests can be used as practical tools to quantify the dynamic process of physical resilience. In addition, physical performance tests can be used as markers of complications related to cancer treatment [47] and as a marker of early death to adapt oncologic therapeutics [51]. Moreover, the objective identification of low physical resilience by physical performance tests can contribute to therapeutic decision-making (e.g., about medication options) and/or to the development of acute care management strategies that reduce complications or complications of treatment, and/or to the development of preventive strategies to decrease the risk of adverse events, such as falls and disabilities in BCS. Furthermore, the objective screening of low physical resilience using the physical performance tests can contribute to the development of early intervention methods capable of improving physical resilience capacity, such as practicing regular physical exercise. For example, it has been shown that physical exercise is a method capable of promoting the increase in physical fitness necessary for improved resilience [52]. In that regard, high resilience is associated with positive outcomes, including successful aging, lower depression, and increased longevity in older adults [53]. Although physical exercise is a promising intervention method for increasing physical function, it is still unclear what magnitude of increase in gait speed would represent a clinical improvement in physical resilience, and thus, further research is needed to investigate this relationship.

In our study, muscle strength measured by the handgrip strength test was lower in the 1–3.9 YAT group when compared to the age-matched women with no history of cancer (i.e., the CT group) (Table 2). Moreover, we observed that the ≥4 YAT group has greater muscle strength when compared to the 1–3.9 YAT group (Table 2). These results, when compared to the results of other physical function tests, suggest that changes in handgrip strength related to the time from the end of cancer treatment do not follow the same pattern as changes in functional tests. One possible explanation is that handgrip strength alone may not be as sensitive as other tests to capture changes in various physiological systems. Reduction in muscle strength is mainly dependent on changes in muscle mass and/or the nervous system [54,55]. Indeed, no reduction in muscle mass has been reported after breast cancer treatments [56]. Furthermore, another important point is that handgrip measures the muscle strength of the upper limbs, which involves a small, specific muscle mass (hands and forearm). In the present study, we did not observe a significant difference in the levels of physical activity between the groups. Maintaining a high level of daily living activities that utilize the upper limbs, such as showering, laundry, and housework, can help to preserve (or restore) upper limb strength after cancer treatment. In this regard, as physical resilience has been understood as a dynamic process of interaction between various physiological systems [11–15], other physical performance tests (e.g., TUG test) can provide more comprehensive information on changes in physiological systems. Therefore, it seems reasonable to assume that physical performance tests, rather than handgrip strength, may be more sensitive to changes in physiological systems and a better tool for assessing physical resilience.

The present study has some limitations. The cross-sectional design limits temporal and causal inferences. Thus, longitudinal studies are still needed to confirm the effectiveness of the physical function tests as a tool to identify the low physical resilience in BCS women. Another limitation is related to the relatively small number of participants given the nature of the study, which may have weakened the statistical analysis. However, physical function was assessed by objective tests, all measurements were carried out by the same experienced evaluator, and the physical function tests evaluated, except for the SPPB test, showed

high reproducibility and reliability reported in the scientific literature [35,57,58], therefore reducing the bias. Finally, this study did not assess any biological marker associated with the level of physiological reserve; however, it is well established in the scientific literature that physical function tests are indicators of physiological reserve [23,42,43].

## 4. Methods

### 4.1. Design

An analytical cross-sectional study was conducted to determine whether objective physical performance tests are effective tools for identifying physical resilience capacity in BCS. To develop the present study, we recruited patients who had completed breast cancer treatment (BCS; n = 146) and who were followed up at a cancer center as well as women with the same characteristics (control women; n = 69) from a neighborhood association, both from the city of Uberaba-MG (Brazil). BCS were invited to participate in the study during routine visits to the cancer treatment center. All data were collected between June 2017 and June 2018.

After verbal acceptance and signing an informed consent form, patients were directed to a reserved room and questioned about their physical activity level, menopause time, diseases, and medications. Clinical data (staging and treatment performed) were self-reported and then checked by hospital staff based on analyzing the patient's medical records. After that, physical function and anthropometric evaluations were conducted. The assessments were carried out in this order: physical activity level questionnaire, anthropometric assessments, balance test, handgrip strength, 4 m walk test, timed up and go test and five-times-sit-to-stand test. After all the evaluations, the BSC groups were divided according to the time after the end of breast cancer treatment: (Recent: <1 year; Intermediate: from 1 to 3.9 years; Late: ≥4 years). Therefore, based on the division criteria, the patients were divided and allocated into four groups: control [(CT: n = 69; Age: 58.5 (56.6–60.4)]; <1 year after the end of treatment [(<1 YAT: n = 60; Age: 57.1 (54.9–59.3)]; 1–3.9 years after the end of treatment [(1–3.9 YAT: n = 45; Age: 55.9 (53.2–58.6)]; and ≥4 years after the end of treatment (≥4 YAT: n = 41; Age: 59.4 (56.9–61.9)]. All data were collected in a reserved room at the cancer treatment center (BCS).

### 4.2. Participants

BCS, who were patients at a cancer treatment center in the city, were invited to participate in this study. Patients of all treatment modalities (surgery, radiotherapy, chemotherapy, or combinations) were included in the research. The inclusion criteria included the following: the patients must have completed breast cancer treatment for breast cancer staging between I and III (early breast cancer) and have no functional limitations to perform the physical tests. Patients who had previously undergone another type of cancer treatment were excluded from the study. The CT consisted of women who had no functional limitations to perform the physical tests and who had not undergone any type of cancer treatment. Then, the University Ethics Committee on the Use of Human Subjects approved the study (number CAAE: 82691818.0.0000.5154).

### 4.3. Level of Physical Activity

The evaluation of the level of physical activity was assessed by the International Physical Activity Questionnaire short form (IPAQ-SF) [59]. Volunteers were considered active if they reported ≥150 min·wk$^{-1}$ moderate-intensity physical activity (3–5.9 METs), ≥75 min·wk$^{-1}$ of vigorous-intensity physical activity (≥6 METs), or a combination of moderate- and vigorous-intensity physical activity to achieve a total energy expenditure of ≥500–1000 MET·min·wk$^{-1}$ [60].

The results of the physical activity levels are presented in Table 1. The "active (%)" shows the percentage of people who meet the physical activity cut-off (more than 150 min of mild physical activity per week and/or more than 75 min of vigorous physical activity per week). The "moderate physical activity (%)" represents the percentage of people who

engaged in moderate physical activity, regardless of the total amount of physical activity they performed. The "vigorous physical activity (%)" indicates the percentage of people who participated in vigorous physical activity, regardless of the overall amount of physical activity they performed.

### 4.4. Anthropometric Measurements

Body mass was measured using a 100 g precision digital scale (G-life®, São Paulo, SP, Brazil). Height was measured using an inextensible tape measure (10 mm accuracy) attached to the wall. Body mass index (BMI) was calculated as the ratio between body mass (kg) and height squared (m$^2$) [59].

#### 4.4.1. Grip Strength

Muscle strength is defined as "the amount of force a muscle can produce with a single maximal effort". As a measure of overall muscle strength, grip strength is commonly used in clinical settings [27]. The handgrip strength test was measured using a manual dynamometer (Jamar®, Bolingbrook, IL, USA) to detect muscle strength in the right and left hands. Three measurements were taken on both hands, and the highest value was recorded and used for statistical analysis. In cancer patients, handgrip strength has high reproducibility and reliability [27,61].

#### 4.4.2. Physical Performance

Physical performance is defined as "an objectively measured whole-body function related to mobility" [27]. Physical performance can be evaluated through the walking speed test, chair stand test, short physical performance battery (SPPB) test, and timed up and go test (TUG) [27].

### 4.5. Meter Walk Test (4-MWT)

The usual walking test was evaluated by the time spent walking at a distance of 4 m. To avoid the influence of acceleration and deceleration, 1 m indentation and extension areas were considered in the test area (6 m total). Two measurements were taken, and the shorter time was considered the valid measurement. The 4-MWT has high reproducibility and reliability in clinical populations [(ICC: $\geq$0.90)] [27,62].

### 4.6. Five-Times-Sit-To-Stand Test (FTSST)

The participants were instructed to perform repeated standing and sitting movements from an armless chair (0.45 m height) as fast as possible. The time to complete 5 full repetitions was recoded. The FTSST has excellent reliability and reproducibility in adults with pathologies [(ICC: $\geq$0.90)] [24,63].

### 4.7. Timed up and Go Test (TUG)

The participants were asked to rise from a chair without using their upper limbs, walk three meters, turn around an obstacle, walk back, and sit down on the chair. They received verbal instructions to begin the test. The timing was computed from the voice command to the time they sat back. The TUG is a reliable and valid test for quantifying functional mobility that may also be useful in following clinical change over time [(ICC: $\geq$0.90)] [27,35].

### 4.8. Short Physical Performance Battery (SPPB)

The SPPB consisted of three tests to detect mobility, and the tests were performed in the following order: balance test, 4-MWT, and FTSST. Each test score varied from 0 to 4 points, and the SPPB total score varied from 0 to 12 points (sum of the scores of the three tests) [37].

The balance test consisted of three positions: side-by-side stand, semi-tandem stand, and tandem stand. The score was based on the time spend in each position (10 s maximum).

The 4-MWT test was evaluated by the time walked over a distance of four meters, which the volunteer self-selected the velocity. Two measurements were taken, and the shorter time was considered as the valid measurement.

The FTSST test was evaluated by the time spent in five maximum velocity squats in a chair with the arms folded across the chest. The technique consisted of full sit and stand position, and the volunteer started in the sit position. The SPPB has good reliability and reproducibility [(ICC: $\geq$0.89)] [27,57].

*4.9. Statistical Analysis*

Data distribution was determined using the Kolmogorov–Smirnov test. Levene's test was used to verify the homogeneity. One-way ANOVA and ANCOVA (GLM) were used to compare the parameters between the groups. The confounding variables used in the ANCOVA were age, menopause, smoking, number of medications, BMI, muscle strength, and physical activity level. When a $p < 0.05$ was observed in ANOVA/ANCOVA, a post hoc test (Bonferroni) was performed. The categorical variables were compared using binary logistic regression. Data were presented as mean, confidence interval of 95% (95% CI), or percentage. The significance level was set at 5%.

**5. Conclusions**

Our findings indicate that physical function tests could be used to detect low physical resilience in middle-aged BCS women. From a clinical standpoint, this study could provide a quick, low-cost, and simple technique for assessing low physical resilience in BCS women.

**Author Contributions:** All authors conceived the experiments; F.M.M., G.O.-J., V.C.L., A.A.d.O., M.A.S.C., L.R.M.F.d.S., R.S.N. and F.L.O. were responsible for the interpreting results and for writing the first draft of the manuscript. F.M.M., M.A.S.C., C.O.A., M.V.C.S. and F.L.O. were responsible for editing the manuscript. All authors have read and agreed to the published version of the manuscript.

**Funding:** The Minas Gerais Research Funding Foundation (FAPEMIG- APQ-00957-22), Coordination for the Improvement of Higher Education Personnel (CAPES-001), and National Council for Scientific and Technological Development (CNPq–302560/2021-1).

**Institutional Review Board Statement:** The study was conducted in accordance with the Declaration of Helsinki, and approved by the University Ethics Committee on the Use of Human Subjects approved the study (Number CAAE: 82691818.0.0000.5154).

**Informed Consent Statement:** Informed consent was obtained from all subjects involved in the study.

**Data Availability Statement:** http://bdtd.uftm.edu.br/handle/123456789/1316, accessed on 27 November 2022.

**Conflicts of Interest:** The authors declare no conflict of interest.

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
