# Peer review of "Physical Function Tests Are Potential Tools to Identify Low Physical Resilience in Women after Breast Cancer Treatment"

_muscles, doi:10.3390/muscles2010009_

Round 1

Reviewer 1 Report

This article aim to find a way to identify the weakness of breast cancer women survivors through physical tests. This is a strength point of this work.

The manuscript is clear and with a relevant content. 

In general, cited references are recent and according the subject.

The experimental design is appropriate to test the propose and the results able to reproduce following the methods described.

Conclusions are consistent with the evidence.

Specific notes:

-Introduction could content more information about why is important to test the BCS and the physiological consequences of treatments that justify tests importance.

-line 188 - you say that upper limbs can be affected by daily activities. Why? 

Reviewer 2 Report

Dear Authors,

Thank you for an interesting paper.

I read your paper and came up with some ideas

Please check my comments.

I hope my comments will improve your manuscript.

Best,

Reviewer 3 Report

This paper aims to investigate whether physical function tests may identify low physical resilience in breast cancer survivors.   Comments and suggestions are as follows:

1.     Abstract: Line 26-27.  The authors used 5 physical function measures in this study. Grip strength should be added as a physical function measure.

2.     Introduction: Resilience has been described in the psychosocial literature as the capacity to maintain or regain well-being during or after adversity. The study aimed to investigate whether different physical function tests may identify a low physical resilience in BCS.  The authors compared the differences of physical function performance between 4 groups (control, < 1 YAT, 1- 3.9 YAT, and ≥ 4 YAT).  It seems the authors believe that physical function represents physical resilience in BCS.  Can the authors provide more explanation in the background to explain why the differences of physical function between the control and breast cancer survivors represent lower physical resilience?

         i.            Introduction, 3rd paragraph, line 69-72: The author states “it has shown that the difference between chronological age versus biological age as measured by physical function tests can be a way to quantify an individual's physical resilience capacity”.  Does the authors implied that the control group in this study represent the chronological age, and the BCS groups represent the biological age?

        ii.            The authors did provide explanations in discussion to relate physical function and physical resilience (physiological reserve, energy reserve capacity, etc.).  It might be helpful to use some of the ideas in the introduction.

3.     Methods are placed after Discussion section.  Please change the sequence.

4.     Methods: line 247. It might be better to just use “Grip Strength” as the heading.  Using “Muscle strength” as a heading is not very precise. 

5.     Methods: line 248-251. The authors measured grip strength in both hands.  Please specify the grip strength of which hand was reported in this study.

6.     Methods: The authors mentioned about the excellent reliability and reproducibility of the physical function measures.  Excellent reliability refers to ICC greater than 0.90 (Koo TK, 2017).  Is that true for all 5 physical function measures?

7.     Methods: The authors mentioned about the excellent reliability of the physical function measures.  It might be helpful to mention about validity of these measures because it might help with our understanding of the relationship between physical function and physical resilience.

8.     Discussion: Line 123.  Typos? “…and system central and peripheral nervous”.

9.     Discussion: Line 154-159.  Grammar issue?  “Recently, it has been shown that constructing recovery trajectories after an agent stressor requires several repeated measures of assessments, thus, the physical performance tests present it as a quick-to-manage tool that results in different gait speed responses that can be used as measures of physical function recovery trajectories, as a marker of treatment-related complications and as a marker of early death to adapt oncologic therapeutics”. 

10.  Discussion: Line 200.  “showed high reproducibility and reliability”.  Please refer to point 6 above.  The reliability might not be excellent for all measures.

Koo TK, Li MY. A Guideline of Selecting and Reporting Intraclass Correlation Coefficients for Reliability Research. J Chiropr Med. 2016 Jun;15(2):155-63. doi: 10.1016/j.jcm.2016.02.012. Epub 2016 Mar 31. Erratum in: J Chiropr Med. 2017 Dec;16(4):346. PMID: 27330520; PMCID: PMC4913118.

Round 2

Reviewer 2 Report

Dear Authors,

Thank you for an interesting paper.

I checked the revised version manuscript.

Currently, you politely replied and properly modified it.  

Comprehensively, I feel yours is well documented.

Your manuscript becomes interesting and beneficial for readers.

Best,
